# Bamboo Salt and Triple Therapy Synergistically Inhibit *Helicobacter pylori*-Induced Gastritis In Vivo: A Preliminary Study

**DOI:** 10.3390/ijms232213997

**Published:** 2022-11-13

**Authors:** Tae Ho Lee, Hang Yeon Jeong, Do Yeon An, Haesung Kim, Jeong-Yong Cho, Do Young Hwang, Hyoung Jae Lee, Kyung-Sik Ham, Jae-Hak Moon

**Affiliations:** 1Department of Food Science and Technology, Chonnam National University, 77 Yongbongro, Gwangju 61186, Republic of Korea; 2Research Group of Aging Metabolism, Korea Food Research Institute, Wanju 55365, Republic of Korea; 3Department of Food Biotechnology, Mokpo National University, Mokpo 58554, Republic of Korea

**Keywords:** *Helicobacter pylori*, bamboo salt, triple therapy, antimicrobial, gastritis, anti-inflammatory

## Abstract

*Helicobacter pylori* infections are a major cause of gastrointestinal disorders, including gastric ulcers, gastritis, and gastric cancer. Triple therapy, using two antibiotics and a proton pump inhibitor, is recommended for the treatment of *H. pylori* infections. However, antibiotic resistance in *H. pylori* is an emerging issue. Bamboo salt, a traditional Korean salt made by baking solar sea salt in bamboo barrels, can ameliorate the symptoms of various gastrointestinal diseases. Herein, we compared the anti-*H. pylori* activity of triple therapy (clarithromycin, metronidazole, and omeprazole), solar salt, and bamboo salt in vivo as a preliminary study. Four-week-old C57BL/6 male mice were inoculated for eight weeks with the *H. pylori* Sydney Strain 1 (SS-1) and orally administered triple therapy drugs and salts for five days. The transcript levels of the *H. pylori*-expressed gene *CagA* and inflammatory cytokines *Tnfα* and *Il-1β* significantly decreased in the bamboo salt treated mice than those in the *H. pylori*-infected control group. This effect was further enhanced by using triple therapy and bamboo salt together. Solar salt caused modest inhibition of *H. pylori*-induced inflammation. We also demonstrated the synergistic effects of bamboo salt and triple therapy against *H. pylori*. Thus, bamboo salt may be a potential candidate agent against the treatment of *H. pylori*-associated gastritis.

## 1. Introduction

*Helicobacter pylori* is a Gram-negative spiral bacillus, classified as a class 1 carcinogen by the International Agency for Research on Cancer [1], with an infection rate of >50% worldwide. Moreover, it is closely correlated with the development of several gastrointestinal diseases, such as gastric cancer, gastritis, gastric ulcer, and duodenal ulcer [1,2,3]. After lung cancer, gastric cancer is the second leading cause of cancer-related mortality worldwide [4]. *H. pylori* infections are also associated with numerous extragastric disorders, such as cardiovascular, neurologic, hematologic, and hepatobiliary diseases, as well as diabetes mellitus and various metabolic syndromes [3,5].

*H. pylori* infection is formally assigned as an infection disease in the Internatioanl Classification of Diseases 11th Revision [6]. Although the treatment regimens for *H. pylori* infections have been the diversity, triple therapy, wherein two antibiotics and one proton pump inhibitor (PPI) are administered simultaneously, is considered a standard first-line regimen [7,8,9]. Antibiotics that are used in triple therapy include tetracycline, clarithromycin, metronidazole, and amoxicillin, while gastric acid inhibitors used include esomeprazole, omeprazole, and lansoprazole [7,8,9].

One triple therapy for *H. pylori* eradication involves a mixture of one PPI and two antibiotics (clarithromycin and amoxicillin). Hence, this therapy is also known as PCA [8,10]. However, the success of triple therapy in treating *H. pylori* infections has been decreasing globally of late, owing to an increase in antibiotic resistance in *H. pylori* [9,11,12,13]. To tackle this issue, a quadruple therapy, PCAM, was developed by adding metronidazole for *H. pylori* eradication [10]. Further, side effects, such as interference with iron [14], formation of fundic gland polyps [15], parietal cell hypertrophy, and hyperplasia in the stomach [16], have been reported with long-term PPI intake for treating gastrointestinal diseases. Previous studies on Wistar rats [17] and women [18] have shown that PPI intake for seven days inhibits calcium absorption. Therefore, further research is needed to improve the efficiency of triple therapies develop alternative therapies, and select suitable antibiotics.

Salt is an important seasoning that enhances the taste of food and aids in its preservation. In addition, it is essential for various physiological functions, such as the maintenance of acid˗base balance and osmotic pressure, active transport of substances in cell membranes, production of gastric acid, and regulation of muscles and nerves in the body. The ratio of sodium chloride and minerals in naturally dried solar sea salt varies, depending on the production method, production region, and collection period. It has been reported to have a higher mineral content than that of refined salt [19,20]. Bamboo salt is a traditional Korean salt. It is produced by baking solar sea salt in a bamboo barrel at high temperatures of 600–1100 °C. Bamboo salts can be either white or dark purple, depending on the baking method [21]. It is used as a folk remedy for disinfection and hemostasis and for treating various inflammatory and gastrointestinal disorders [21]. This salt has a strong sulfur odor, like that of a rotten egg, when dissolved in water and eaten. Bamboo contains hydrogen sulfide (H_2_S)-released metal sulfides [22]. H_2_S has been recently recognized as an important endogenous gaseous signaling molecule for regulating organ development and maintaining homeostasis in tissues [22].

Excessive salt intake, however, has been reported to have various adverse effects on health, such as high blood pressure [23], worsening of *H. pylori* infections [24], and increasing the incidence of gastric cancer [25]; the extent of these adverse effects depends on the type of salt consumed, such as refined salt, solar salt, or bamboo salt [26,27,28,29]. Purple bamboo salt has been reported to exhibit therapeutic effects against various diseases of the digestive system, such as gastritis, gastric ulcer, and gastric cancer. In addition, it exhibits various pharmaceutical properties, such as anti-allergic, anti-inflammatory, and anticancer effects [26,27,28,29]. Therefore, we propose that supplementing the triple therapy with bamboo salt could be effective in eradicating *H. pylori* and reducing gastritis.

This study compares the therapeutic effects of triple therapy (administration of clarithromycin, metronidazole, and omeprazole), solar salt, and bamboo salt on *H. pylori*-induced gastritis in vivo as a preliminary study. Therefore, this study used clarithromycin and metronidazole as antibiotics, which were observed to have high antibiotic resistance in triple therapy [8,9], to certify the therapeutic effects of the triple therapy and supplement against *H. pylori* infection.

## 2. Results

### 2.1. Effect of Bamboo Salt on the Eradication of H. pylori Infection and Cytokine Expression

*H. pylori*-infected mice were simultaneously treated with bamboo salt and antibiotics against *H. pylori* to determine the therapeutic effects of bamboo salt. Eradication of *H. pylori* and the anti-inflammatory activity of bamboo salt after five days of treatment because of the complete eradication of *H. pylori* generated at seven days, were analyzed by comparing the relative mRNA expression of the *H. pylori*-specific gene *CagA* and the inflammatory cytokines *Tnfα* and *Il-1β*. As shown in Figure 1A, *CagA* expression was not observed in the triple therapy-treated groups (T, ST, and BST), indicating that all *H. pylori* strains were eradicated by the antibiotics. In addition, the BS group showed a declining trend in *CagA* expression compared to that exhibited by the infected HP group, although this difference was not significant. Both *Tnfa* and *Il-1β* levels were significantly increased in the HP group than that in the control group (Figure 1B,C). While the solar salt treatment (S) had little effect on the reduction in the expression of inflammatory cytokines, the BS group showed a decline in the expression of inflammatory cytokines compared to that in the HP group (Figure 1B,C). When solar and bamboo salts were administered together with triple therapy (ST and BST), the expressed inflammatory cytokine levels were observed to be similar or lower compared to that in the group treated with triple therapy alone (Figure 1B,C). In particular, when bamboo salt was administered together with triple therapy (BST), inflammatory cytokines were observed to be similar to or lower than those in the non-infected group (C, Figure 1B,C). These results suggest that bamboo salt exhibits excellent synergistic effects when used in combination with conventional antibiotics.

### 2.2. Synergistic Effects of Bamboo Salt in Combination with Triple Therapy

To determine the synergistic effect and the concentration-dependent manner of the action of triple therapy with bamboo salt, triple therapy was administered to *H. pylori*-infected mice at 1, 1/2, 1/4, and 1/8 times the existing dose. As shown in Figure 2, *Tnfa* and *Il-1β* expressions decreased in a concentration-dependent manner with triple therapy. Interestingly, in BST, *Tnfa* and *Il-1β* expressions were lower at all concentrations than that in T or ST. Next, we compared the cumulative anti-inflammatory effects of triple therapy and different concentrations of salt. The triple therapy was administered at one-tenth of the existing dose, while salts were administered at twice or half the recommended daily intake. As shown in Figure 3, similar expression trends were observed for both *Tnfa* and *Il-1β*. In the groups administered salts alone (S, BS), treatment with half the concentration of salt (1/2S, 1/2BS) led to a greater decline in the *Tnfa* and *Il-1β* expression compared with that in the groups treated with double the concentration of salt (2S, 2BS), regardless of the type of salt. In particular, *Tnfα* and *Il-1β* expression in the 1/2S solar salt-administered group was similar to that in the HP group. However, *Tnfα* and *Il-1β* expressions were significantly higher in the 2S group than those in the HP group, suggesting that a high concentration of solar salt may aggravate gastritis. Although there was no significant difference, the anti-inflammatory effect showed an increasing trend in the BST group compared with that in the T group. In particular, when double the concentration of bamboo salt was administered with triple therapy, the mRNA expression of inflammatory cytokines was observed to be similar to that in the non-infected group (C). These results suggest that bamboo salt enhanced the antibacterial and anti-inflammatory effects of antibiotics.

### 2.3. Effect of Bamboo Salt on the Gastric Tissue Histology of H. pylori-Infected Mice

We compared the histology of H&E-stained mouse gastric tissues to confirm the therapeutic effect of bamboo salt on gastric mucosal damage caused by *H. pylori* infection (Figure 4). In H&E staining, the cell nucleus and cytoplasm are stained purple and pink, respectively; and the nuclei of normal cells were observed to be circular. In contrast, the cell nucleus becomes rhombic or oval due to cell necrosis in the inflamed tissue of the *H. pylori*-infected mice. The degree of inhibition of inflammation in the salt treated groups was compared based on these histological characteristics. The level of inflammation in terms of histology was compared for the following five groups: C, HP, T, half the concentration of bamboo salt group (1/2BS), and half the concentration of bamboo salt + triple therapy group (1/2BST). As shown in Figure 4, significant inflammation was observed in the HP group (Figure 4B) compared with that in the non-infected group (Figure 4A). A significant inflammatory treatment effect was observed in the three groups (Figure 4C–E) administered with the triple therapy and bamboo salt compared with that in the infected group (Figure 4B); however, no significant difference was found among these three treatment groups.

## 3. Discussion

In this study, we determined the therapeutic effect of mineral-rich solar salt [19,20] and bamboo salt on gastrointestinal disorders [27,28] in an attempt to develop alternative therapies to overcome the decreasing success of triple therapy in eradicating antibiotic-resistant *H. pylori* infections [8,9,11,12]. To evaluate the effectiveness of using salt as an adjuvant for treating *H. pylori* infection and its anti-inflammatory effects in vivo, salts and triple therapy were administered simultaneously. The anti-inflammatory effect of triple therapy increased in a concentration-dependent manner regardless of the salt administration (Figure 2). Although the low salt concentration showed enhanced anti-inflammatory effects in *H. pylori* infection when bamboo and solar sea salt administered singly, there were no significantly different anti-inflammatory effects exhibited by the salt concentration in *H. pylori* infection when salts and triple therapy were simultaneously administered (Figure 3).

CagA is one of the most widely known virulence factors of *H. pylori* encoded by the *CagA* gene and is strongly associated with inflammation and gastric cancer pathogenesis [1]. In this study, the degree of *H. pylori* eradication was evaluated by analyzing the *CagA* expression using qPCR. *CagA* expression (Figure 1A) showed no change in the S group compared with that in the HP group. Although there was no significant difference, mice administered BS alone showed a decreasing trend in the expression compared to that in the HP group. Unfortunately, it was difficult to accurately determine the antibacterial activity of bamboo salt because of the low number of mice used in the experiment and large deviation. This study considered only five days of treatment with bamboo salt. Therefore, it is necessary to increase the number of study animals and the treatment duration in future studies to accurately evaluate the antibacterial activity of bamboo salts.

The anti-inflammatory effect of bamboo salt was evaluated by comparing the expression of inflammatory cytokines *Tnfα* and *Il-1β* across different groups. A significantly lower expression of the inflammatory cytokines was observed in the BS group than that in the HP group. Therefore, bamboo salt exhibits anti-inflammatory effects in *H. pylori*-infected mice. In addition, when bamboo salt and triple therapy were administered simultaneously (BST), a decreasing trend was observed in the mRNA expression of *Tnfα* and *Il-1β* compared with that in the T group at the same concentration. Upon examining the synergistic effect of various concentrations of antibiotics and bamboo salt, mice administered both bamboo salt and triple therapy showed the lowest expression of inflammatory cytokines. Side effects of antibiotic administration have recently begun to gain attention. Recent studies have shown that epidermal growth factor receptor signaling associated with gastric inflammation and carcinogenesis remains activated even after antibiotic-induced eradication of *H. pylori* [30]. Further, antibiotic treatment promotes inflammation through the translocation of native intestinal bacteria [30]. Although additional evaluation is needed, our results suggest that side effects caused by antibiotics can be reduced when bamboo salt is administered together with antibiotics. However, adverse effects, such as high blood pressure [23] due to excessive salt intake, have been reported and thus regular intake of excessive salt should be prevented.

Solar sea salt and bamboo salt might play a role in attenuating oxidative stress and inflammation induced by dietary salt containing only NaCl [31,32]. K, Mg, and Ca are the main minerals in solar sea salt and bamboo salt. However, in this study, we confirmed that bamboo salt was more effective in reducing gastric damage than solar sea salt is, even though the former contains less minerals compared to that in the latter. However, the exact mechanism through which bamboo salt exhibits various physiological activities has not yet been elucidated. Notably, bamboo salt is more alkaline (approximately pH 11) than solar sea salt (approximately pH 9.0) [27,33]. Alkaline mineral water has been reported to alleviate ethanol-induced gastric ulcers in mice by inhibiting the pepsin activity and increase of the levels of prostaglandin E-2 (PGE2) and heat shock protein 70 (HSP70) [34]. Therefore, both bamboo and solar sea salts may be inhibited gastric damage induced by *H. pylori* owing to their alkaline properties. Metal sulfides (such as MgS, NaSH, Na_2_S, etc.) present in bamboo salt release H_2_S, which inhibits the gastric damage induced by aspirin and ethanol by activating the ATP-sensitive potassium channel [35,36,37]. Therefore, H_2_S might act as an excellent bioactive compound of bamboo salt with a protective effect against gastric damage [27,38]. In addition, bamboo salt has a higher content of hydroxyl groups, and hence, it shows a higher antioxidant activity than that exhibited by solar sea salt and general dietary salts [33,39]. Therefore, we hypothesize that the antibacterial and inflammatory therapeutic effects of bamboo salt against *H. pylori* are due to its alkaline property and the generation of H_2_S and hydroxyl groups.

In summary, we evaluated the therapeutic effects of bamboo salt on gastritis. In addition to the anti-inflammatory activity of bamboo salt, we showed that it exhibits antibacterial activity against *H. pylori*. Thus, we report a new biological activity of bamboo salt and its potential as an adjuvant in the treatment of *H. pylori* infections.

## 4. Materials and Methods

### 4.1. Chemicals

Solar salt was purchased from the Seongchang Salt Farm (Shinan, Korea), and bamboo salt was provided by Amicogen Co., Ltd. (Jinju, Korea). Clarithromycin, metronidazole, and omeprazole were purchased from TCI Chemical Industry (Tokyo, Japan). All other chemicals and solvents were of analytical grade unless specified otherwise.

### 4.2. H. pylori Strain and Culture Conditions

Mouse-adapted *H. pylori* Sydney Strain-1 (SS-1) was obtained from the Korean Culture Center of Microorganisms (Seoul, Korea) and cultured on Columbia agar or in broth medium (MB cell, Seoul, Korea) containing 5% horse serum (Gibco, Gaithersburg, MD, USA). The cultures were incubated at 37 °C in a 10% CO_2_ incubator (MCO175, Sanyo, Osaka, Japan) and sub-cultured every 72 h [40,41]. Culture purity was assessed regularly.

### 4.3. H. pylori Infection of Animals

All experimental procedures were approved by the Institutional Animal Care and Use Committee of Chonnam National University (no. CNU IACUC-YB-2019-31). Four-week-old C57BL/6 male mice were purchased from Samtako Bio Korea (Osan, Korea). The mice were reared in an environmentally controlled animal facility operating on a 12:12 h dark/light cycle at 20 ± 1 °C and 55 ± 5% humidity with ad libitum access to water and standard laboratory chow (Harlan Rodent diet, 2018S, Samtako Bio Korea) [41].

Three mice per group were inoculated with *H. pylori* SS-1, which can effectively colonize mouse gastric mucosa [42]. The mice were orally administered 200 µL of Columbia broth containing 10^8^ colony forming units of *H. pylori* daily for eight weeks using a Zonde needle. The negative control group consisted of uninfected mice administered the same volume of fresh Columbia broth.

### 4.4. Treatment of Animals with Bamboo Salt and Triple Therapy Drugs after H. pylori Infection

Following eight weeks of *H. pylori* inoculation, the infected mice were administered 200 μL of bamboo salt in water once daily for five days [41,43]. Groups of mice were also administered either solar salt or triple therapy. The mice were divided into seven experimental groups (n = 3): control group (C, uninfected), *H. pylori*-infected group (HP, negative control), *H. pylori* infection + triple therapy treatment group (T), *H. pylori* infection + solar salt treatment group (S), *H. pylori* infection + solar salt + triple therapy treatment group (ST), *H. pylori* infection + bamboo salt treatment group (BS), and *H. pylori* infection + bamboo salt + triple therapy treatment group (BST). Triple therapy comprised omeprazole (700 μg/kg body weight, standard dose PPI for humans), metronidazole (16.7 mg/kg body weight, 1 g a day for humans), and clarithromycin (16.7 mg/kg body weight, 1 g a day for humans), as mentioned by Jung et al. [9]. The concentration of solar salt and bamboo salt administered was 166.6 mg/kg body weight (two times of 83.3 mg, the recommended daily intake of salt). After five days of treatment, blood was drawn from the abdominal aorta of the mice under light anesthesia via the respiratory system using isoflurane. The pyloric antrum of the stomach was dissected and harvested for quantitative polymerase chain reaction (qPCR) and histological analysis. All samples were stored at −80 °C until further use.

To determine the synergistic effect of the bamboo salt and different triple therapy concentrations, the following four concentrations of triple therapy were administered: existing concentration (metronidazole and clarithromycin: 16.7 mg/kg; omeprazole: 700 μg/kg), one-half, one-quarter, and one-eighth of the existing concentration. Solar and bamboo salts were administered at the aforementioned concentrations (166.6 mg/kg body weight).

We also evaluated the synergistic effect of bamboo salt and low˗dose triple therapy (one-tenth of the existing dose). Bamboo and solar salts were administered at half (83.3 mg/kg body weight) and double (333.2 mg/kg body weight) the standard concentration (166.6 mg/kg body weight). The mice were euthanized after five days of treatment, and the samples were harvested as aforementioned.

### 4.5. Gene Expression Analysis

Total RNA was isolated from mouse gastric tissues using TRI Reagent^®^ (Molecular Research Center, Cincinnati, OH, USA). cDNA was synthesized using the ReverTra Ace^®^ qPCR RT kit (Toyobo, Osaka, Japan), and qPCR amplification was performed using Rotor-Gene Q (QIAGEN, Hilden, Germany). The primer sequences are listed in Table 1. The mRNA expression levels were normalized to those of the internal control gene mouse ribosomal protein, Large, P0 (*Rplp0*) using the comparative threshold cycle method [44].

### 4.6. Histological Examination

Gastric tissues were fixed in 4% (*w/v*) paraformaldehyde in phosphate-buffered saline (pH 7.4) for 24 h, dehydrated in a graded ethanol series (70%, 80%, 90%, 95%, and 100%), cleared in xylene, embedded in paraffin, and sectioned into 5-μm-thick slices. Serial sections were stained with hematoxylin and eosin (H&E) and examined microscopically to visualize pathological lesions in the gastric mucosa [45].

### 4.7. Statistical Analysis

Data are presented as mean ± standard deviation and were determined using the Statistical Package for Social Sciences (SPSS, IBM, Armonk, NY, USA) version 20.0. Statistically significant differences were determined using one-way analysis of variance followed by Tukey–Kramer and Student’s *t*-tests. *p* < 0.05 was considered significant.

## Figures and Tables

**Figure 1 ijms-23-13997-f001:**
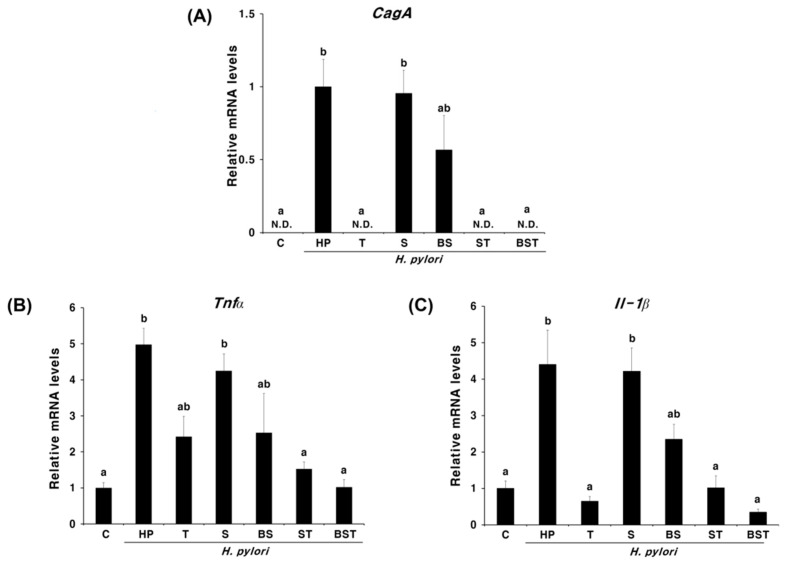
Expression levels of *CagA* (**A**), *Tnfα* (**B**), and *Il-1β* (**C**) in the pylorus tissue of *Helicobacter pylori*-infected and control mice administered triple therapy, solar salt, and/or bamboo salt. Different letters indicate a significant difference (*p* < 0.05) ascertained via the Turkey˗Kramer test. C, control group (uninfected); HP, infected group (negative control); T, triple therapy treatment group (positive control); S, solar salt treatment group; BS, bamboo salt treatment group; ST, solar salt + triple therapy treatment group; BST, bamboo salt + triple therapy treatment group.

**Figure 2 ijms-23-13997-f002:**
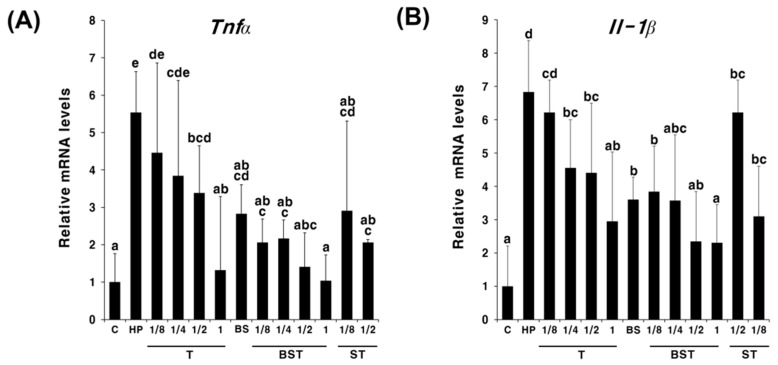
Expression levels of *Tnfα* (**A**) and *Il-1β* mRNA (**B**) in the pylorus tissue of *Helicobacter pylori*-infected and control mice administered triple therapy and different concentrations of solar and bamboo salts. Different letters indicate a significant difference (*p* < 0.05) ascertained via the Turkey-Kramer test. C, control group (uninfected); HP, infected group (negative control); A, triple therapy treatment group (positive control); BS, bamboo salt treatment group; ST, solar salt + triple therapy treatment group; BST, bamboo salt + triple therapy treatment group. 1, existing concentration of triple therapy; 1/2, half of existing concentration of triple therapy; 1/4, quarter of existing concentration of triple therapy; 1/8, one-eighth of existing concentration of triple therapy. Salt-treated groups were administered equal concentrations of solar salt or bamboo salt.

**Figure 3 ijms-23-13997-f003:**
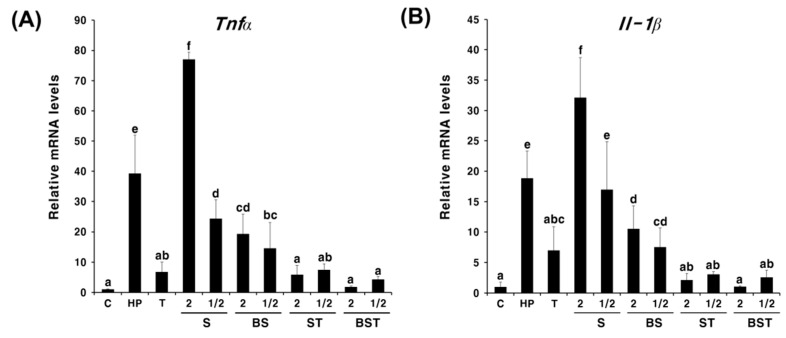
Expression levels of *Tnfα* (**A**) and *Il-1β* mRNA (**B**) in the pylorus tissue of *Helicobacter pylori*-infected and control mice administered different concentrations of solar and bamboo. Different letters indicate a significant difference (*p* < 0.05) ascertained via the Turkey-Kramer test. C, control group (uninfected); HP, infected group (negative control); T, triple therapy treatment group (positive control); S, solar salt treatment group; BS, bamboo salt treatment group; ST, solar salt + triple therapy treatment group; BST, bamboo salt + triple therapy treatment group. 2, Double of the recommended daily intake of salt; 1/2, half of the recommended daily intake of salt. Triple therapy-treated groups were administered the same dose of triple therapy.

**Figure 4 ijms-23-13997-f004:**
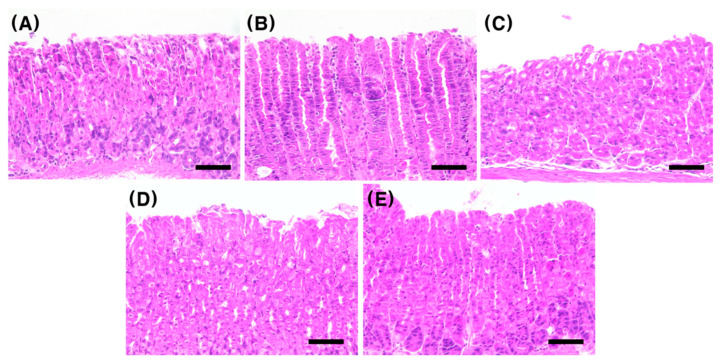
Histological analysis of mouse gastric tissue by H&E staining (original magnification ×400). (**A**) uninfected control group; (**B**) infected HP group; (**C**) infected and treated T group; (**D**) infected and treated 1/2BS group; (**E**) infected and treated 1/2BST group. 1/2, half of the recommended daily intake of salt. Scale bar = 50 μm.

**Table 1 ijms-23-13997-t001:** Sequences of primers used for quantitative polymerase chain reaction.

Gene	Sequence	
	Forward	Reverse
*Rplp0*	GTGCTGATGGGCAAGAAC	AGGTCCTCCTTGGTGAAC
*Tnfα*	CGAGTGACAAGCCTGTAGCC	AGCTGCTCCTCCACTTGGT
*Il-1β*	ATGAGAGCATCCAGCTTCAA	TGAAGGAAAAGAAGGTGCTC
*CagA*	CCGATCGATCCGAAATTTTA	CGTTCGGATTTGATTCCCTA

*Rplp0*, ribosomal protein, Large, P0; *Tnfα*, tumor necrosis factor alpha; *Il-1β*, interleukin-1 beta; *CagA*, cytotoxin-associated gene A.

## Data Availability

Not applicable.

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
