# Peer review of "Bamboo Salt and Triple Therapy Synergistically Inhibit Helicobacter pylori-Induced Gastritis In Vivo: A Preliminary Study"

_ijms, 2022, doi:10.3390/ijms232213997_

Round 1

Reviewer 1 Report

The authors of the paper entitled "Bamboo salt and triple therapy synergistically inhibit Helicobacter pylori-induced gastritis in vivo”" aimed to investigate the effect of 5days treatment of triple therapy with addition of bamboo and solar salts.

This topic is very interesting and the novelty of the paper is good, however I have some major concerns which make the work unsuitable for publication in its current form

major concerns:

1.   The number of the studied animals per group is very low which make me doubt the significance of statistical analysis. In order to maintain good, believable and statistically significant results the number of the animals in each group should be at least 5 whereas in this study n=3. With such a number, the power of statistical tests is too low to obtain conclusive results

2.       Authors do not justify why they choose these particular drugs among others for triple therapy, while in the introduction they mention several antibiotics and drugs that reduce gastric acid secretion that are used in triple therapy in humans

3.       The discussion section is too short, the authors should elaborate on the topic of Helicobacter pylori-induced gastritis and emphasize the importance of presented research

minor concerns:

4.   Authors should describe how the animals were euthanised during the present research

5.   Did the mice show any symptoms of gastritis?

6.      How were established the doses of the drugs and salts? Are such doses of drugs used in humans?

7.       Why the therapy lasted only 5 days?

Author Response

We appreciate the reviewer’s comments and advice to improve the quality of our manuscript.

(major concerns)

1. The number of the studied animals per group is very low which make me doubt the significance of statistical analysis. In order to maintain good, believable and statistically significant results the number of the animals in each group should be at least 5 whereas in this study n=3. With such a number, the power of statistical tests is too low to obtain conclusive results

<Answer> Thanks for the reviewer’s comment. We also agree with the pointing out. However, because there are many experimental groups and difficult to apply a large number of animals in our present laboratory, we have used the smallest number (n = 3) of animals. We wish the reviewer’s understands on this point.

2. Authors do not justify why they choose these particular drugs among others for triple therapy, while in the introduction they mention several antibiotics and drugs that reduce gastric acid secretion that are used in triple therapy in humans

<Answer> We revised and added the sentences (lines 45-50 and lines 83-86) of the introduction to explain why we chose clarithromycin and metronidazole as antibiotics. The references also changed to a recent version.

3. The discussion section is too short, the authors should elaborate on the topic of Helicobacter pylori-induced gastritis and emphasize the importance of presented research

<Answer> We revised and added the discussion (lines 187-193 and lines 223-242). The references (21, 30, 31, 33, and 34) were also added.

(minor concerns)

4. Authors should describe how the animals were euthanised during the present research

<Answer> We revised it as the verse “via the respiratory system using isoflurane” in the sentence of line 287 of Materials and Methods.

5. Did the mice show any symptoms of gastritis?

<Answer> We could not confirm the symptoms of H. pylori infection occurred in mice. We thought it may because of the short-duration infection of H. pylori.

6. How were established the doses of the drugs and salts? Are such doses of drugs used in humans?

<Answer> The doses of drugs were explained at lines 282-286 of Materials and Methods. The dose of drugs is recommended for humans, and the dose of dietary salt is also recommended as daily salt intake by WHO.

7. Why the therapy lasted only 5 days?

<Answer> We described this at lines 91-92 in Result’s section. We used stomach tissues treated during the five days for comparing gastricis-related factors among the groups because of the complete eradication of H. pylori generated at seven days.

Reviewer 2 Report

The quality of this article is not so good that it cannot be published on this journal

Author Response

Thank you for the review of our manuscript.

Reviewer 3 Report

An in vitro study, to compare with the in vivo one, would have been interesting. 

Author Response

(The authors gave the same response as above.)

Reviewer 4 Report

The idea of inventing new drugs or supplements to increase H.pylori eradication rate and / or reduce gastritis independently from H.pylori eradication is great, however the quality of the study or at least its description is unsatisfactory.

The introduction gives many outdated information and cites very old articles. The authors are affiliated with Food Science and Food Biotechnology Depatrments, it would be really worth for them to consult the data from the introduction with gastroenterologist. For example nowadays it is no longer believed that H.pylori cause dermatologic, neurologic or metabolic syndromes, neither it has anything to do with head and neck.

What did the authors mean writing that H.pylori treatment regimen depends from H.pylori status? Status is either positive and negative and the regimens depend from something else. Triple therapy is no longer gold standard – here the authors cite the articles from the years 2007-2014. The current Maastricht VI guidelines were published in 2021 and the previous Maastricht V guidelines in 2017. Still triple therapy is one of the options.

The authors cite 3 articles from 2014 to prove the growing resistance of H.pylori to antibiotics while the newest data are from 2021 doi.org/10.1136/gutjnl-2020-321372 and 2022: doi: 10.3390/ijerph19116921.

The interference with iron absorption and polyps growth has been proved for longstanding PPI intake not for the 14 day intake while H.pylori eradication.

What kind of salt is baked to have Bamboo salt – is if sea salt baked? Why is it important to bake it in bamboo? Maybe baking it in the oven would be the same? Does baking in Bamboo cause migration of some substances from bamboo into the salt?

It is not clear to me how many groups there were. In line 157 authors write about 11, in line 240 bout 7. When I count all the regiment of different doses of antibiotics and all salts concentrations – it will be at least 22 groups. If in each group there were 3 mice – how many was sacrificed together?

What is the aim of different doses of antibiotics? I do not know the data about direct anti-inflammatory effect of antibiotics – only depending from the effect on eradication. What does existing dose (line108) mean – is it a standard dose? If yes- in all mice treated with standard dose the eradication was successful. I have found no information about the success rate in lower concentrations of antibiotics.

The number of the animals in the study groups wat low and the authors admit, that the deviastion of results was large (line181). It would be better to   have less groups  (for example only one concentration of substances) and more animals in group.

Line 92

Lower reduction in my opinion means less effect. And I have understood the authors proved more effect so greater reduction.

Author Response

We appreciate the reviewer’s comments and advice to improve the quality of our manuscript.

â–¶ The idea of inventing new drugs or supplements to increase H.pylori eradication rate and / or reduce gastritis independently from H.pylori eradication is great, however the quality of the study or at least its description is unsatisfactory.

<Answer> We agree that the quality of the study is not very high. However, this study compared the anti-inflammatory effect between the bamboo salt and triple therapy group. These results suggest that bamboo salt and triple therapy synergistically inhibited H. pylori-induced gastritis in vivo. Therefore, we submitted our manuscript as COMMUNICATION.

â–¶ The introduction gives many outdated information and cites very old articles. The authors are affiliated with Food Science and Food Biotechnology Departments, it would be really worth for them to consult the data from the introduction with gastroenterologist. For example nowadays it is no longer believed that H.pylori cause dermatologic, neurologic or metabolic syndromes, neither it has anything to do with head and neck.

<Answer> We revised this with citation of recent published papers at lines 30-56.

â–¶ What did the authors mean writing that H.pylori treatment regimen depends from H.pylori status? Status is either positive and negative and the regimens depend from something else. Triple therapy is no longer gold standard – here the authors cite the articles from the years 2007-2014. The current Maastricht VI guidelines were published in 2021 and the previous Maastricht V guidelines in 2017. Still triple therapy is one of the options.

<Answer> We revised and cited recent version published in 2022 by Malfertheriner et. (lines 38-39).

â–¶ The authors cite 3 articles from 2014 to prove the growing resistance of H.pylori to antibiotics while the newest data are from 2021 doi.org/10.1136/gutjnl-2020-321372 and 2022: doi: 10.3390/ijerph19116921.

<Answer> We revised this with citation of recent published papers (lines 45-56).

â–¶ The interference with iron absorption and polyps growth has been proved for longstanding PPI intake not for the 14 day intake while H. pylori eradication.

<Answer> We revised this in lines 50-54.

â–¶ What kind of salt is baked to have Bamboo salt – is if sea salt baked? Why is it important to bake it in bamboo? Maybe baking it in the oven would be the same? Does baking in Bamboo cause migration of some substances from bamboo into the salt?

<Answer> Bamboo salt (BS) is made from solar sea salt containing various minerals in a bamboo barrel by baking at high temperature of 600–1100 oC and has a strong sulfur odor like rotten egg when it is dissolved in water and eaten. That is, various sulfur-containing compounds such as OCS, CS2, MgS, Na2S, NaHS, CaS, FeS, K2S, etc. are contained in BS and then hydrogen sulfide (H2S) is released when BS is dissolved in water. However, other salts did not contain or in very small amount H2S. Our experimental data as below are also indicated that sulfate in solar sea salt and carbon in bamboo may play an important role in the formation of H2S-releasing compounds during BS manufacturing. We are preparing another paper on these results.

<Our experimental data>

BS exhibited overwhelmingly higher H2S-releasing content than other salts [mineral-rich solar sea salt (MRS), mineral-deficient salt (MDS, refined salt), MRS treated by heat without bamboo (roasted salt, RS)] (Figure 1). In particular, H2S released by RS, which is made from mineral rich solar sea salt (MRS) treated by heat without bamboo, was very less. Therefore, bamboo might be considered as an important ingredient on the formation of H2S-releasing compounds during BS manufacturing.

Several studies have been reported that sulfate is reduced with carbons at high temperature of 600–1100 oC (1,2). When solar sea salt treated by heat with bamboo is manufactured, solar sea salts containing minerals such as Mg, Ca, K, etc. as well as sulfate are generally used as feedstocks and processed at roasting temperature (3,4). H2S-releasing compounds may be produced by sulfate reduction with carbon during the high-temperature roasting process. That is, the sulfate ions would be reduced to sulfide ions through the following reactions.

SO42-(s) + 2C (s, bamboo) ® S2-(s) + 2CO2 (g) or

SO42-(s) + 4C (s, bamboo) ® S2- (s) + 4CO (g)

The produced sulfide ions can form metal sulfides such as MgS and Na2S that can release H2S in water and acidic aqueous solution, respectively.

MgS (s) + 2H2O (l) ® Mg(OH)2 (s) + H2S (g)

Na2S (s) + H2O (s) ® 2Na+ (aq) +  HS- (aq) + OH- (aq)

The bisulfide can generate H2S by combining with H+ in aqueous environment.

To study the main factors responsible for the formation of H2S-releasing compounds during roasted salt manufacturing, NaCl was treated by heat in the presence of sulfate sources (MgSO4 and Na2SO4) with/without bamboo. H2S was released from NaCl treated by heat in the presence of sulfate sources (MgSO4 and Na2SO4) and bamboo (Figure 2).  In particular, the amount of H2S released from NaCl treated by heat in the presence of bamboo increased depending on the sulfate concentration (below 5% of material). The amount of H2S released from NaCl treated by heat with bamboo and without sulfate source was very small, which may be attributed to sulfur-containing compounds present in bamboo. However, H2S was not released from NaCl treated by heat without sulfate source or bamboo.

References

  1. Oduro H. et al. Evidence of magnetic isotope effects during thermochemical sulfate reduction. Proc. Natl. Acad. Sci. USA 108, 17635–17638 (2011).
  2. Kloužek J. et al. Redox equilibria of sulphur in glass melts. Ceramics-Silikáty 50, 134–139 (2006).
  3. Gao TC et al. Mineral-rich solar sea salt generates less oxidative stress in rats than mineral-deficient salt. Food Sci. Biotechnol. 23, 951-956 (2014).
  4. Zhao X. et al. Chemical properties and in vivo gastric protective effects of bamboo salt. Food Sci. Biotechnol. 23, 895–902 (2014).

â–¶ It is not clear to me how many groups there were. In line 157 authors write about 11, in line 240 bout 7. When I count all the regiment of different doses of antibiotics and all salts concentrations – it will be at least 22 groups. If in each group there were 3 mice – how many was sacrificed together?

<Answer> We revised it in line 168. We confirmed the level of inflammation in terms of histology for the following five groups (C, HP, T, 1/2BS, and 1/2BST).

â–¶ What is the aim of different doses of antibiotics? I do not know the data about direct anti-inflammatory effect of antibiotics – only depending from the effect on eradication. What does existing dose (line108) mean – is it a standard dose? If yes- in all mice treated with standard dose the eradication was successful. I have found no information about the success rate in lower concentrations of antibiotics.

<Answer> We confirmed the synergistic effect and the concentration-dependent manner of the action of triple therapy with bamboo salt. The anti-inflammatory effect of triple therapy increased in a concentration-dependent manner. And the anti-inflammatory effect of triple therapy with bamboo salt also increased in a concentration-dependent manner. (Fig. 2).

The mean of the existing dose is a mixture of metronidazole (16.7 mg/kg), clarithromycin (16.7 mg/kg), and omeprazole (700 μg/kg) as a positive control (T) (in line 293).

â–¶ The number of the animals in the study groups wat low and the authors admit, that the deviation of results was large (line181). It would be better to have less groups (for example only one concentration of substances) and more animals in group.

<Answer> Thanks for the reviewer’s comment. We also agree with the pointing out. However, because there are many experimental groups and difficult to apply a large number of animals in our present laboratory, we have used the smallest number (n = 3) of animals. We wish the reviewer’s understands on this point.

â–¶Line 92, Lower reduction in my opinion means less effect. And I have understood the authors proved more effect so greater reduction.

<Answer> We revised it to line 102-105.

Previous sentence “When salts were administered together with triple therapy (ST and BST), similar or lower reductions in inflammatory cytokine levels were observed compared with that in the group treated with triple therapy alone (Fig. 1B and 1C).” à Revised sentence “When solar and bamboo salts were administered together with triple therapy (ST and BST), the expressed inflammatory cytokine levels were observed to be similar or lower compared to that in the group treated with triple therapy alone (Fig. 1B and 1C).”

Round 2

Reviewer 1 Report

Although the topic of the manuscript "Bamboo salt and triple therapy synergistically inhibit Helicobacter pylori-induced gastritis in vivo”" is very interesting and the novelty of the paper is good, the number of the animals in the studied groups is not enough to obtain conclusive statistical analysis. The power of the statistical tests that authors used for the statistical analysis for n=3 is very low. Therefore, all of the observed changes are only suppositions and the presented results does not fully support the conclusions. This experiment should be continued on a larger groups of animals (at least 5 in one group). The presented study can only be treated as a preliminary data. All my other comments were taken into account and the manuscript was revised according to my guidelines. Therefore, it has to be highlighted that this work need to be continued and the presented paper can be published as a communication in the International Journal of Molecular Sciences. I also suggest to add to the title that this is only a preliminary study. 

Author Response

Thank you very much for the reviewer’s comments and advice to improve the quality of our manuscript.

Comments and Suggestions for Authors

Although the topic of the manuscript "Bamboo salt and triple therapy synergistically inhibit Helicobacter pylori-induced gastritis in vivo” is very interesting and the novelty of the paper is good, the number of the animals in the studied groups is not enough to obtain conclusive statistical analysis. The power of the statistical tests that authors used for the statistical analysis for n=3 is very low. Therefore, all of the observed changes are only suppositions and the presented results does not fully support the conclusions. This experiment should be continued on a larger groups of animals (at least 5 in one group). The presented study can only be treated as a preliminary data. All my other comments were taken into account and the manuscript was revised according to my guidelines. Therefore, it has to be highlighted that this work need to be continued and the presented paper can be published as a communication in the International Journal of Molecular Sciences. I also suggest to add to the title that this is only a preliminary study.

(Answer) To highlight a preliminary study, we added a pharse of a preliminary study in the sentence in the title (line 3), abstract (line 17), and introduction (line 83).

Reviewer 4 Report

The article has much improved although some statements are still not comprehensible to me, such as line 39:" treatment regimens for H. pylori infections depend on the infection status" and not all the literature that I suggested was cited. 

Author Response

Thank you very much for the reviewer’s comments and advice to improve the quality of our manuscript.

Comments and Suggestions for Authors

The article has much improved although some statements are still not comprehensible to me, such as line 39:" treatment regimens for H. pylori infections depend on the infection status" and not all the literature that I suggested was cited.

(Answer) We revised “Although the treatment regimens for H. pylori infections depend on the infection status” to “Although the treatment regiments for H. pylori infections have been the diversity” in line 39. And we also cited the reference (13, 2022: doi: 10.3390/ijerph19116921) in line 48.
